# Approximate Bayesian Neural Operators: Uncertainty Quantification for Parametric PDEs

**Emilia Magnani**                                          *emilia.magnani@uni-tuebingen.de*
*Tübingen AI Center, University of Tübingen*

**Nicholas Krämer**[*]                                          *pekra@dtu.dk*
*Technical University of Denmark*

**Runa Eschenhagen**[*]                                          *re393@cam.ac.uk*
*University of Cambridge*

**Lorenzo Rosasco**                                          *lrosasco@mit.edu*
*MaLGa - DIBRIS, University of Genova, Istituto Italiano di Tecnologia*

**Philipp Hennig**                                          *philipp.hennig@uni-tuebingen.de*
*Tübingen AI Center, University of Tübingen*

**Reviewed on OpenReview:** *https://openreview.net/forum?id=6WvIkYsMA8*

## Abstract

Neural operators are a type of deep architecture that learns to solve (i.e. learns the nonlinear solution operator of) partial differential equations (PDEs). The current state of the art for these models does not provide explicit uncertainty quantification. This is arguably even more of a problem for this kind of tasks than elsewhere in machine learning, because the dynamical systems typically described by PDEs often exhibit subtle, multiscale structure that makes errors hard to spot by humans. In this work, we first provide a mathematically detailed Bayesian formulation of the "shallow" (linear) version of neural operators in the formalism of Gaussian processes. We then extend this analytic treatment to general deep neural operators—specifically, graph neural operators—using approximate methods from Bayesian deep learning, enabling them to incorporate uncertainty quantification. As a result, our approach is able to identify cases, and provide structured uncertainty estimates, where the neural operator fails to predict well.

## 1 Introduction

Neural operators (Kovachki et al., 2023; Li et al., 2020b; 2021a; 2020a; 2021b) are deep learning architectures designed for reconstruction problems related to partial differential equations (PDEs). They approximate mappings between infinite-dimensional vector spaces of functions, such that – once trained – solutions of entire families of parametric PDEs can be represented by a single neural network. However, the learning process is subject to several sources of uncertainty, which can result in a potentially significant prediction error because of the nonlinear – and often nonintuitive – interactions of different stages of the approximation. The goal of this paper is to develop methods for estimating this error at a practically acceptable computational cost. This kind of functionality is urgently needed in this domain: Due to the intricate and often not intuitive nature of the dynamical systems described by PDEs, it can be hard for the human eye to detect prediction errors, even when they are large.

In this paper, we address this gap by developing an approximate Bayesian framework for neural operators – from a theoretical, and a computational point of view. We begin with a brief review of neural operators.

---

[*]Work mostly done at the University of Tübingen.

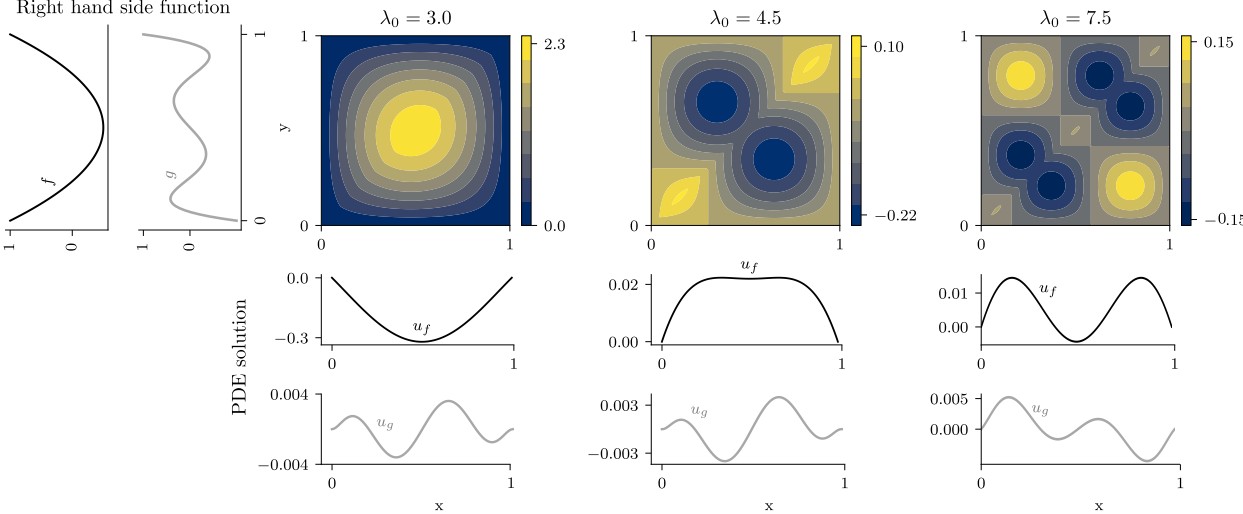

Figure 1: Green's functions in Equation (6) for different values of $\lambda_0 = \{3, 4.5, 7.5\}$. On the left, right-hand-side functions $f$, $g$ for the PDE in Equation (5) and respective solutions $u_f$, $u_g$ for the correspondent $\lambda_0$-value, computed through Equation (4).

Then, using linear, parametric PDEs as guiding examples, we show how their "shallow" (single-layer) base case allows for an analytic Bayesian treatment using the formalism of Gaussian processes (Rasmussen & Williams (2006)). This linear case, while primarily of theoretical interest, provides valuable insights and aims to make this model class more accessible to the Bayesian machine learning community. We then extend the theoretical analysis to the nonlinear deep case. Here, analytic treatments are no longer possible, so we fall back on approximations developed for Bayesian deep learning. Specifically, we focus on Laplace approximations (MacKay, 1992) which are easy to add post-hoc even to pretrained networks, and add only moderate computational cost relative to deep training without uncertainty quantification (Daxberger et al., 2021). Our experiments in Section 5 demonstrate that the resulting method effectively captures structure in the predictive error of graph neural operators, both in the over- and under-sampled regime. In Section 2 we discuss some theoretical background, and develop a probabilistic framework for neural operators in Section 3. We discuss related work in Section 4.

## 2 Background

In this section, we examine how neural operators approximate solution operators for parametric PDEs through functional observations. If we fix one input of the solution operator, neural operators can be understood as effectively inverting the differential operator associated with the PDE. In this framework, the process of learning the operator becomes equivalent to reconstructing the Green's function, reducing the problem to a task of function approximation. This perspective, developed in Section 2.1, forms the basis for the Bayesian approach developed in Section 3.1. Subsequently, in Section 2.2, we outline the iterative structure of neural operators, their training procedure, and how they relate to Green's functions.

### 2.1 PDEs And Green's Function

One of the main fields of applications of neural operators are families of parametric PDEs of the form

$$
\begin{aligned}
(\mathcal{L}_\lambda u)(x) &= f(x), & x \in D \\
u(x) &= 0, & x \in \partial D
\end{aligned}
\tag{1}
$$

for some sufficiently well-behaved, bounded domain $D \subset \mathbb{R}^d$ with boundary $\partial D$ (e.g. open, bounded $D$ with Lipschitz boundary $\partial D$), where $U \ni u \colon D \to \mathbb{R}$, $F \ni f \colon D \to \mathbb{R}$, $\lambda \in \Lambda$, with $U$, $F$ and $\Lambda$ appropriate

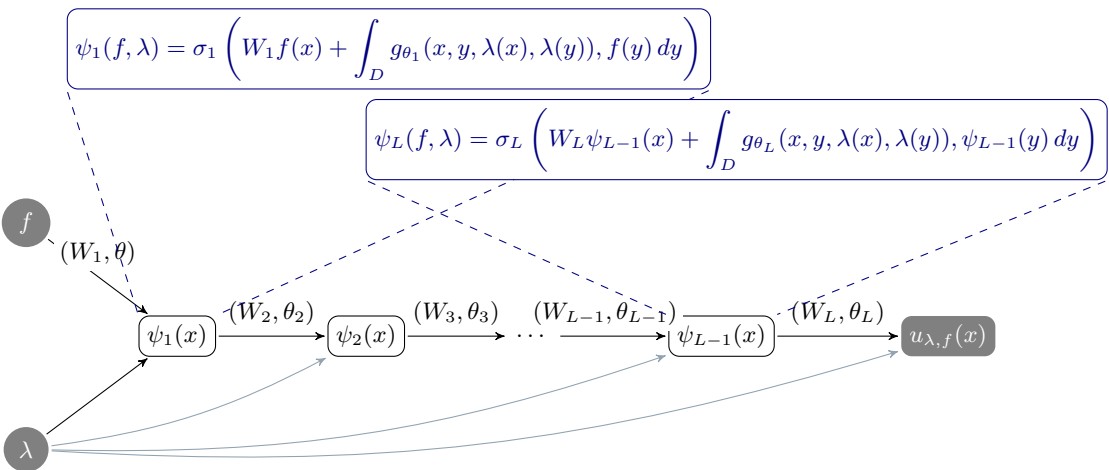

Figure 2: Neural operator architecture $NO_\Theta$. Each layer $l$ computes a new function $\psi_l$, that contains the neural network $g_\theta$ in the integrand. Layer parameters are shown on the corresponding arrows.

function spaces. The precise nature of those function spaces is not important for the remainder of this work. The function $\lambda$ parametrises the differential operator $\mathcal{L}_\lambda$.

Equation (1) defines a solution operator

$$\mathcal{H} \colon \Lambda \times F \to U, \quad (\lambda, f) \mapsto u_{\lambda,f} \tag{2}$$

in the sense that $\mathcal{H}(\lambda, f)(x) = u_{\lambda,f}(x)$ solves the PDE for the given functions $\lambda$ and $f$. Even though the PDE is linear, $\mathcal{H}$ is (possibly highly) nonlinear. The operator $\mathcal{H}$ is a map between function spaces. The idea behind neural operators is to approximate the operator $\mathcal{H}$ with a single neural network trained on function observations $\{f_i, u_i\}_{i=1}^N$. Thus, instead of approximating the solution of the PDE for only a fixed $f$ or $\lambda$, neural operators directly infer the operator $\mathcal{H}$. Numerically, the functions $f$ and $u$ are observed on a discretisation grid of the function domains.

In this subsection we are interested in the particular case where $\lambda$ is fixed, so the solution operator can be written as

$$\mathcal{G} \colon f \mapsto u. \tag{3}$$

If the differential operator $\mathcal{L}_\lambda$ is linear, the map $\mathcal{G}$ inherits that linearity. Considering the operator in Equation (3) is an important step to understand the learning process of neural operators. In fact, observe how $\mathcal{G}$ is the inverse of the operator $\mathcal{L}_\lambda$. In this simplified case where $\lambda$ is fixed, the neural operator is therefore learning an operator, $\mathcal{G}$, through function observations $\{f_i, u_i\}_{i=1}^N$ that derive from the action of its inverse. In other words, during training, the neural operator is implicitly learning to invert the differential operator $\mathcal{L}_\lambda$. In particular, in the case where the differential operator is linear and admits a Green's function $G$, the solution of Equation (1) can be expressed through integration with the kernel $G$

$$u_\lambda(x) = \int_D G_\lambda(x, y) f(y) \, \mathrm{d}y. \tag{4}$$

Hence, learning the operator $\mathcal{G}$ is here equivalent to learn the function $G_\lambda$, which means that an operator-learning task can be reduced to that of function-reconstruction.

In the general analysis of linear PDEs (we refer to e.g. Evans (2010) for background on PDEs), the Green's function $G_\lambda(x, y)$ represents the impulse response of the linear operator $\mathcal{L}_\lambda$, that is $\mathcal{L}_\lambda(G_\lambda)(\cdot, y) = \delta(\cdot - y)$ for $y \in D$, where $\delta$ denotes the Dirac delta distribution. Despite $\mathcal{L}_\lambda$ being linear, the Green's function itself can be nonlinear in in either arguments. To visualize these concepts, we consider the one-dimensional boundary value problem

$$\begin{aligned} \left( -\Delta - \lambda_0^2 \operatorname{Id} \right) u(x) &= f(x), \quad x \in [0, 1], \\ u(0) = u(1) &= 0, \end{aligned} \tag{5}$$

where in this case $\lambda_0 \in \mathbb{R}$ is a (scalar) parameter, and Id is the identity operator. It's Green's function, for $\lambda_0 \neq n\pi \ \forall n \in \mathbb{N}$, is given by

$$G_{\lambda_0}(x, y) := \frac{A + B}{\lambda_0 \sin(\lambda_0)} \tag{6}$$

where we abbreviated

$$A := H(y - x)\sin(\lambda_0 x)\sin(\lambda_0(1 - y)) \tag{7}$$
$$B := H(x - y)\sin(\lambda_0(1 - x))\sin(\lambda_0 y), \tag{8}$$

and $H$ denotes the Heaviside step function. Details of the Green's function derivation are in Appendix A. Equation (5) relates to Equation (1) in the sense that the differential operator $\mathcal{L}_{\lambda_0} = (-\Delta - \lambda_0^2 \operatorname{Id})$ is parametrised by $\lambda_0$. Figure 1 shows examples of Green's functions $G_{\lambda_0}$ for different values of $\lambda_0$, along with solutions computed via Equation (4).

## 2.2 Overview of Neural Operators

Neural operators are neural-network-based architectures designed to approximate the general solution operator $\mathcal{H}$ defined in Equation (2). Before introducing our Bayesian framework, we briefly review their structure. A more thorough explanation of what follows can be found in the work by Kovachki et al. (2023); Li et al. (2020b; 2021a; 2020a; 2021b).

Let $g_\theta : D \times D \times \mathbb{R} \times \mathbb{R} \to \mathbb{R}$ be a neural network with parameters $\theta$. Define the neural operator $\text{NO}_\Theta$ as a composition of $L \in \mathbb{N}$ layers

$$\begin{aligned} \text{NO}_\Theta : \Lambda \times F &\to U, \\ (\lambda, f) &\mapsto (\psi_L \circ \psi_{L-1} \circ \ldots \circ \psi_1)(\lambda, f), \end{aligned} \tag{9}$$

where each layer

$$\psi_\ell : \Phi \to \Phi, \quad \ell = \{1, \ldots L\}, \tag{10}$$

is defined as a composition of (i) integrating the output of the previous layer against $g_{\theta_\ell}$, and (ii) combining the integral with a linear component and an activation function $\sigma$,

$$\psi_\ell(h)(x) = \sigma\left(W_\ell h(x) + \int_D g_{\theta_\ell}(x, y, \lambda(x), \lambda(y))h(y) \, \mathrm{d}y\right). \tag{11}$$

The space $\Phi$ in Equation (10) is a vector space of real-valued functions on $D$, and the final layer of the neural operator maps into $U$, so $\psi_L : \Phi \to U$. In Equation (11), $W_\ell$ is a learnable linear operator (represented by a matrix after discretization), and $g_{\theta_\ell}$ is the integral kernel in the $\ell$-th layer. In practice, the integral cannot be computed in closed-form and a suitable quadrature formula needs to be employed (which turns the integral into a weighted sum of evaluations of the integrand; see e.g. Davis & Rabinowitz (2007)). The parameter set $\Theta$ of $\text{NO}_\Theta$ is $\Theta = \{\theta_\ell \cup W_\ell\}_{\ell=1}^L$. Loosely speaking, one can think of this construction as a deep neural network ($\text{NO}_\Theta$) that iteratively approximates the solution $u_{\lambda, f}$ with linear transformations $W_\ell$ and nonlinear activation functions $\sigma$, and at every iteration (layer) employs another neural network ($g_{\theta_\ell}$). For a visualisation of $\text{NO}_\Theta$ see Figure 2. Although the figure shows $\lambda$ entering in each layer, in practice the kernel may encode $\lambda$ in an initial "lifting" layer while using a final "projection" layer to map the function output back to the physical domain.

Note how $\text{NO}_\Theta$ approximates an operator. While, technically speaking, this means that its training and test set consist of functions, in the numerical computation, these functions need to be observed on some grid. Nonetheless, neural operators are resolution-agnostic: their architecture does not depend on a particular discretization grid, and they can be applied to different resolutions without retraining. Let $\{\lambda_1, ..., \lambda_N\} \times \{f_1, ..., f_M\}$ be a set of training inputs, each of which shall be observed on some mesh $\mathbb{X} := \{x_1, ..., x_K\}$. In total, that makes $NK \times MK = NMK^2$ training inputs. Without loss of generality, and for the sake of simple notation, assume that the solution of the PDE and the respective inputs are observed on the same

mesh $\mathbb{X}$. Thus, we observe $NM$ solutions $u_{11}, ..., u_{NM}$, i.e. $NMK$ training outputs – one set of evaluations at $\mathbb{X}$ for each solution $u_{nm}$ associated with $(\lambda_n, f_m)$, $n = 1, ..., N$, $m = 1, ..., M$. Each of these outputs is a function that maps from $D$ to $\mathbb{R}$, thus $u_{nm}(\mathbb{X}) \in \mathbb{R}^K$. The relation between inputs and outputs is

$$u_{nm} = \mathcal{H}(\lambda_n, f_m) \approx \text{NO}_\Theta(\lambda_n, f_m). \tag{12}$$

While this equation is between functions, once discretised, it becomes an equation between vectors. To be able to optimise the parameters, we introduce the loss function

$$\mathcal{L} : \mathbb{R}^K \times \mathbb{R}^K \to [0, \infty). \tag{13}$$

The network parameters $\Theta$ are then computed by (approximately) solving the minimisation problem

$$\Theta^* = \arg\min_\Theta \sum_{n,m} \mathcal{L}(u_{nm}(\mathbb{X}), \text{NO}_\Theta(\lambda_n, f_m)(\mathbb{X})), \tag{14}$$

where we used the above vectorised notation. This minimisation can be carried out with any of the optimisers popular in deep learning (see e.g. (Le et al., 2011)). Note that by approximating directly the solution operator $\mathcal{H}$, $\text{NO}_\Theta$ simultaneously learns the entire family of PDEs parametrised by $f, \lambda$ without the need of re-training the network for a new $\lambda$ or $f$. Considering that these new inputs samples can be out of distribution cases, which are notoriously harder to predict (Hendrycks & Gimpel, 2017), it is even more important to introduce uncertainty quantification for these architectures.

### 2.2.1 The One-Layer (Shallow) Case

A special, *shallow* version of the neural operator arises by setting $L = 1$, $\sigma \equiv \text{Id}$, and $W_1 = 0$, with $\lambda \equiv \lambda_0^2$ fixed. In this simplified scenario, we can focus on the operator $\mathcal{G} : f \mapsto u$ from Equation (3), yielding

$$\text{NO}_\Theta(f) = \text{NO}_\theta^{\text{shallow}}(f) = \int_D g_\theta(x, y) f(y) \, \mathrm{d}y. \tag{15}$$

where $g_\theta := g_{\theta_1}$ is now the only learned integral kernel. If $g_\theta$ is a sufficiently accurate approximation of the Green's function, such as $G_{\lambda_0}$ in Equation (6), then Equation (15) essentially recovers the classical solution integral $\int_D G_{\lambda_0}(x, y) f(y) \, \mathrm{d}y$. Hence, the structure of neural operators in its one-layer (shallow) case is inspired by the Green's solution formula for linear PDEs (Equation (4)). In the next section, we provide a Gaussian process–based probabilistic perspective on this one-layer operator, which lays the groundwork for a more general Bayesian treatment of multi-layer (deep) neural operators.

## 3 Method

In this section, we develop the Bayesian probabilistic framework for neural operators. Section 3.1 focuses on the special case of a one-layer (shallow) network, where we can leverage Gaussian process regression to obtain an analytic non-parametric Bayesian treatment. This setting provides not just a useable algorithm, but also an important conceptual base-case that is not prominently discussed in previous works on neural operators (including non-Bayesian ones). In Section 3.2, this "shallow" treatment is extended to the deep setting using a linearisation in form of the Laplace approximation, which again provides a Gaussian posterior distribution, albeit an approximate one.

### 3.1 Bayesian Neural Operators In The Shallow Case With Gaussian Processes

We begin with the shallow neural operator $\text{NO}_\theta^{\text{shallow}}$ introduced in Equation (15). In particular, we consider the *linear* PDE in Equation (5). In this setting, the PDE's solution operator $\mathcal{G} : f \mapsto u$ can be approximated via $\text{NO}_\theta^{\text{shallow}}(f) = \int_D g_\theta(x, y) f(y) \, \mathrm{d}y$, where $g_\theta(x, y)$ plays the role of the Green's function $G(x, y)$. Since the considered linear PDE admits an analytic Green's function $G$ (see Equation (6)), and since the only parameters of $\text{NO}_\Theta$ are the ones of the neural network $g_\theta$ (i.e. $\Theta = \theta$), learning $\mathcal{G}$ reduces to learning the function $G : \mathbb{R}^2 \to \mathbb{R}$.

**Formulating the Problem as GP Regression.** In contrast to conventional GP regression, instead of directly observing values of $G$, we only observe integrals of $G$ against various input functions. Specifically, for each training input function $f_n$, we observe

$$u_n(x) = \int_D G(x,y) f_n(y) \, \mathrm{d}y, \quad n = 1, \ldots, N.$$

Define the integral operator $\mathcal{A}_f = \mathcal{A}$ acting on $G$ as

$$(\mathcal{A}G)(\cdot) = \int_D G(\cdot, y) f(y) \, \mathrm{d}y.$$

Because $\mathcal{A}$ is a linear operator in $G$, a Gaussian likelihood involving these observations (including the limit case of noise-free observations) ensures conjugacy when we place a GP prior over $G$. Concretely, suppose

$$G \sim \mathcal{GP}(\mu, k_\theta), \quad u \mid G \sim \mathcal{N}(\mathcal{A}G, \sigma^2),$$

where $\mu \colon \mathbb{R}^2 \to \mathbb{R}$ is the prior mean function and $k_\theta \colon \mathbb{R}^2 \times \mathbb{R}^2 \to \mathbb{R}$ is the covariance kernel parameterized by $\theta$. Because both the prior and the likelihood are Gaussian with a *linear* observation model, the posterior over $G$ remains Gaussian (Tanskanen et al., 2020; Longi et al., 2020).

**Posterior Mean and Covariance.** The resulting posterior distribution over $G$ is again a GP with mean and covariance:

$$\mathbb{E}[G] = \mu + \mathcal{A}^* k_\theta \left( \mathcal{A}\mathcal{A}^* k_\theta + \sigma^2 \right)^{-1} (u - \mathcal{A}\mu),$$

$$\mathrm{Cov}(G) = k_\theta - \mathcal{A}^* k_\theta \left( \mathcal{A}\mathcal{A}^* k_\theta + \sigma^2 \right)^{-1} \mathcal{A} k_\theta,$$

$$(16)$$

where $\mathcal{A}^*$ is the adjoint operator of $\mathcal{A}$. To see why, note that we have a standard linear–Gaussian model $(u_i = \mathcal{A}_{f_i} G + \sigma^2)$, where observations $u_i$ are obtained via the linear operator $\mathcal{A}$ acting on $G$ which yields a closed-form GP posterior (Rasmussen & Williams, 2006).

**Interpretation and Extensions.** This Gaussian posterior enables the usual suite of GP-based inference tools, such as computing uncertainty estimates on predictions and drawing posterior samples. Moreover, prior domain knowledge about Green's functions (e.g., symmetry $G(x,y) = G(y,x)$) can be incorporated into the kernel $k_\theta$ (Duvenaud, 2014). Since the solution $u$ is a linear function of $G$, once $G$ is learned, any new input function $f^*$ can be mapped to a distribution over solutions $u^*$. That is, even in this simple "shallow" scenario, we obtain a probabilistic estimate over the solution operator of th PDE. In Section 5.1, we illustrate the use of this GP approach on Equation (5).

### 3.2 From Gaussian Processes to Neural Networks: Last-Layer Laplace Approximation

While the GP-based approach from Section 3.1 provides an *exact* Bayesian treatment for the shallow (one-layer) operator, it does not directly extend to *deep* neural operators, whose non-linearities break the linear–Gaussian framework. Instead, we can adopt approximate inference methods from Bayesian deep learning to approximate the posterior distribution $p(\Theta \mid \mathcal{D})$, where $\mathcal{D} = \{\lambda_n, f_m, u_{nm}\}$, $n = 1, \ldots, N$, $m = 1, \ldots, M$ are the training data, and $\Theta$ are the network parameters. In particular, we use the *Laplace approximation*, a relatively simple yet powerful approach to approximate the parameter's posterior distribution with a Gaussian (MacKay, 1992; Blundell et al., 2015).

**Predictive Distribution.** To make predictions at test inputs $(\lambda_*, f_*)$, we need the predictive distribution

$$p(u_* \mid \mathrm{NO}_\Theta(\lambda_*, f_*), \mathcal{D}) \approx \int p(u_* \mid \mathrm{NO}_\Theta(\lambda_*, f_*)) q(\Theta) \, \mathrm{d}\Theta \tag{17}$$

where $q(\Theta) \approx p(\Theta \mid \mathcal{D})$ is the approximate posterior. In general, computing this predictive distribution requires further approximation; for example, a local linearization of the neural network (Immer et al., 2020) yields a Gaussian predictive distribution under a Gaussian likelihood. A simpler yet often effective alternative is to focus on a *last-layer* Laplace approximation, as we describe below.

**Laplace Approximation.** The Laplace approximation for neural networks is built around the maximum a-posteriori (MAP) estimate of $\Theta$. Denote the regularized training loss as

$$\mathcal{L}(\mathcal{D}; \Theta) = r(\Theta) + \sum_{n,m} \ell(\lambda_n, f_m, u_{nm}, \Theta), \tag{18}$$

where $\ell$ corresponds to the negative log-likelihood $-\log p(u_{nm} \mid \text{NO}_\Theta(\lambda_n, f_m))$ and $r(\Theta)$ is the negative log-prior. Then the MAP weights are

$$\Theta_{\text{MAP}} = \arg\min_\Theta \mathcal{L}(\mathcal{D}; \Theta).$$

Near $\Theta_{\text{MAP}}$, we approximate $\mathcal{L}(\mathcal{D}; \Theta)$ via a second-order Taylor expansion:

$$\Theta_{\text{MAP}} = \arg\min_\Theta \mathcal{L}(\mathcal{D}; \Theta) = \arg\min_\Theta \left( r(\Theta) + \sum_{n,m} \ell(\lambda_n, f_m, u_{nm}, \Theta) \right), \tag{19}$$

where the first-order term disappears at $\Theta_{\text{MAP}}$. Then the posterior approximation $q(\Theta)$ can be identified as a Gaussian centered at $\Theta_{\text{MAP}}$, with a covariance corresponding to the local curvature:

$$q(\Theta) := \mathcal{N}(\Theta \mid \Theta_{\text{MAP}}, (\nabla_\Theta^2 \mathcal{L}(\mathcal{D}; \Theta)|_{\Theta_{\text{MAP}}})^{-1}). \tag{20}$$

Hence, the approximate posterior is Gaussian, centered at $\Theta_{\text{MAP}}$, with a covariance given by the inverse Hessian of the loss at that point.

**Practical Advantages.** Standard training of neural networks already identifies the local optimum $\Theta_{\text{MAP}}$. Thus, the main additional cost is computing the Hessian $\nabla_\Theta^2 \mathcal{L}(\mathcal{D}; \Theta)$ at $\Theta_{\text{MAP}}$, once. Moreover, this procedure can be done *post hoc* on a pre-trained network, which implies that uncertainty quantification in the form of a Laplace approximation comes only at a very small computational overhead while also preserving the predictive power of the maximum a posteriori estimate.

**Last-Layer Laplace in Neural Operators.** To apply the Laplace method efficiently, one typically decomposes the network into a fixed feature map corresponding to the first $L-1$ layers and a last linear layer (Snoek et al., 2015). In the graph neural operator by Li et al. (2020b), the last layer is linear in its weights. This linearity ensures that a Gaussian posterior on the last-layer weights induces a Gaussian distribution over the operator outputs. Hence, for a Gaussian likelihood the predictive distribution in Equation (17) can be computed in closed form by using the approximate posterior $q(\Theta)$. Note that this predictive distribution is equivalent to the one of a GP regression problem (Khan et al., 2019). Conceptually, this connects the shallow GP approach to the deep case, although we are now not approximating the posterior over the parameters of the Green function, but over the weights of the last layer.

Recent work (Kristiadi et al., 2020; Daxberger et al., 2021) has shown that this approach achieves competitive performance on many common uncertainty quantification benchmarks compared to more recent alternatives – despite the low computational overhead. In Section 5, we demonstrate that the same methodology can be effectively combined with graph neural operator architectures to provide predictive uncertainties for PDE solutions.

**Mesh independence.** Neural operators are resolution-agnostic: they do not depend on a specific spatial discretization and can be applied to different grids without retraining. In graph neural operators, for example, the message-passing scheme does not require a fixed-resolution grid, making them flexible in handling various discretizations. This property, although not a primary focus of this work, is inherited by the last-layer Laplace approximation because it models uncertainty in parameter space rather than in the discretization domain.

## 4 Related work

The interplay of (parametric) partial differential equation models (see Cohen & DeVore (2015) for a review) and deep learning has rapidly gained momentum in recent years. Broadly speaking, there are two approaches:

learning the solution of a given PDE on the one hand, and learning the parameter-to-solution operator of a family of parametric PDEs on the other hand.

Conventional numerical PDE solvers (e.g. Ames (2014)) and physics-informed neural networks (PINNs) (Raissi et al., 2019; Sirignano & Spiliopoulos, 2018; Zhu et al., 2019) fall into the first category. In PINNs, the PDE solution is modelled as a neural network. The differential equation is then translated into an appropriate loss function, and an approximate PDE solution emerges from automatic differentiation and numerical optimisation. While the physics-informed neural network formulation extends naturally to PDE inverse problems (Raissi et al., 2019; Zhu et al., 2019), it brings with it some practical issues like hyperparameter-sensitivity and complicated loss landscapes (Wang et al., 2021; Sun et al., 2020; Krishnapriyan et al., 2021). PINNs also need to be retrained once the parametrisation of the PDE ($\lambda$ or $f$ in the discussion above) changes.

As described in Section 2.2, neural operators do not face this issue because they learn the parameter-to-solution operator of a family of parametric PDEs (recall Equation (2)). Conceptualised by Lu et al. (2021), brought to the limelight by Bhattacharya et al. (2021); Nelsen & Stuart (2021); Li et al. (2020b;a; 2021a;b); Patel et al. (2021); Duvall et al. (2021); Kovachki et al. (2023), neural operators have since been extended into a range of architectures. These include graph neural operators (Li et al., 2020a), Fourier neural operators (Li et al., 2021a), multi-wavelet neural operators (Gupta et al., 2021), and physics-informed neural operators (Li et al., 2024), which integrate data and PDE constraints to simultaneously leverage observed data and governing equations. For a comprehensive overview of neural operator architectures, we refer to Azizzadenesheli et al. (2024). Work on universal approximation results for neural operator architectures include Kovachki et al. (2023; 2021); Lanthaler et al. (2023). In this work, we focus on *graph neural operators* (Li et al., 2020b) for our experimental study.

Despite these advances, *uncertainty quantification* remains relatively underexplored in neural operators. Recent efforts include Kumar et al. (2024), which combine a wavelet neural operator with a Gaussian process prior by optimizing hyperparameters via the marginal likelihood; Zou et al. (2024), who propose a Bayesian extension of DeepONets; and Garg & Chakraborty (2022), who employ variational inference. In parallel, kernel- and GP-based operator-learning approaches (Batlle et al., 2024a; Magnani et al., 2024; Chen et al., 2021; Batlle et al., 2024b; Chen et al., 2024) address function-space mappings, and Boullé & Townsend (2022) focus specifically on learning Green's functions for PDEs. Our approach differs by providing an exact, GP-based formulation for the *shallow* (one-layer) operator under linear PDEs, and a *post-hoc* last-layer Laplace approximation for deep graph neural operators. While Magnani et al. (2024) also apply Laplace approximations in the context of neural operators, their focus is on Fourier neural operators rather than graph-based architectures. Beyond neural operators, Laplace approximations have also been used in other neural PDE solvers, such as deep Galerkin methods (Beltran et al., 2024) and PINNs (Izzatullah et al., 2022). Uncertainty quantification is particularly critical in low-data regimes, where generating training data requires expensive numerical simulations. Bayesian methods provide a principled way to assess predictive reliability under such constraints.

Outside the PDE context, approximate Bayesian treatments for neural networks include variational inference (Graves, 2011; Blundell et al., 2015; Khan et al., 2018; Zhang et al., 2018), Markov Chain Monte Carlo (Neal, 1996; Welling & Teh, 2011; Zhang et al., 2020), and heuristic methods (Gal & Ghahramani, 2016; Maddox et al., 2019). Most such approaches require either re-training or specialized sampling mechanisms, which can be computationally expensive and may alter the optimization process. The Laplace approximation (MacKay, 1992; Kristiadi et al., 2020; Daxberger et al., 2021) circumvents these downsides by approximating the posterior *around* a standard (non-Bayesian) pre-trained computation. This property makes it especially appealing for neural operators, where training is typically time- and resource-intensive.

## 5 Experiments

In this section, we apply the theoretical framework from Section 3 to construct Bayesian neural operators that provide uncertainty estimates. We begin with the *shallow* case, leveraging the exact Gaussian process formulation of Section 3.1, and then proceed to the deep setting. By replicating experiments from Li et al. (2020b), we show that we can effectively detect wrong predictions. In Section 5.3, we evaluate our method

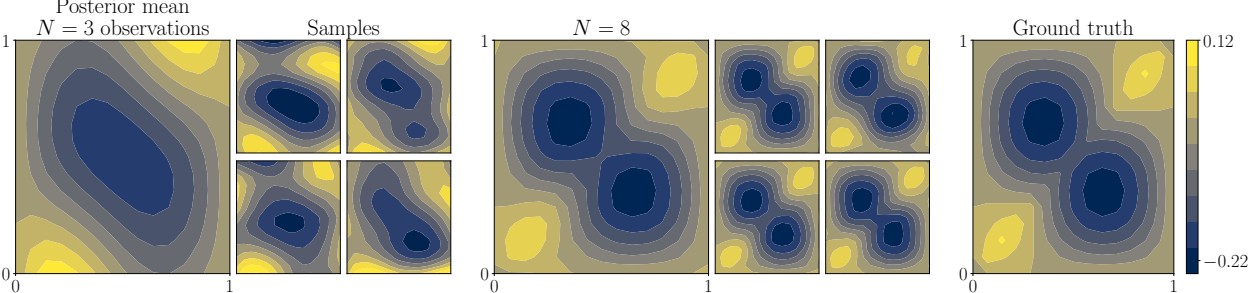

Figure 3: Posterior distribution on $G_{\lambda_0}$ for $\lambda_0 = 4.5$ (and ground truth) after $N = 3, 8$ observations $\{f_i\}_{i=1}^N$ with $f_i$ shifted Legendre polynomials. The samples show the approximation's variance, which decreases when $N$ increases.

on a benchmark for PDEs and compare its performance with other widely used uncertainty quantification methods.

## 5.1 Uncertainty Quantification in the Shallow Case with GP regression

We first consider the boundary-value problem from Equation (5) with a fixed parameter $\lambda_0 \in \mathbb{R}$. Since this linear PDE admits a Green's function $G \colon \mathbb{R}^2 \to \mathbb{R}$, learning the solution operator $\mathcal{G} \colon f \mapsto u$ reduces to estimating $G$ from integral observations

$$\left\{ \left( f_i, \ u_i = \int_D G(\cdot, y) \, f_i(y) \, \mathrm{d}y \right) \right\}_{i=1}^N .$$

Numerically, each right-hand side function $f_i$ and the corresponding solution $u_i$ are observed on an evenly spaced grid $\mathbb{X} = \{x_1, \ldots x_K\}$. As training functions $\{f_i\}$, $i = 1, \ldots, N$, we use the first $N$ shifted Legendre polynomials, evaluated on $\mathbb{X}$. We then place a Gaussian prior $G \sim \mathcal{GP}(\mu, k)$ with a zero mean function $\mu$ and a kernel function $k \colon \mathbb{R}^2 \times \mathbb{R}^2 \to \mathbb{R}$ that factorizes into the product $k((x_0, x_1), (y_0, y_1)) = k_1(x_0, y_0) k_2(x_1, y_1)$ where $k_1$ and $k_2$ are Matérn kernels with parameter $\nu = 2.5$ . The integral operator $\mathcal{A}$ in Equation (16) is computed via numerical quadrature.

Figure 3 shows samples from the resulting posterior over $G$ for $\lambda_0 = 4.5$ when $N = 3$ and $N = 8$. Samples from the posterior are used to visualize the posterior variance. For $N = 3$, the posterior variance is large, indicating a high degree of uncertainty. As $N$ increases to 8, the posterior variance diminishes significantly, yielding a closer approximation to the true Green's function. Since learning $G$ corresponds to learning the inverse of the differential operator in Equation (5), the posterior distribution over $G$ can be leveraged to obtain both an approximation of the solution and an associated error estimate for a new PDE with right-hand side function $f^*$.

## 5.2 Uncertainty Quantification in the Deep Case: Darcy flow

We now showcase the role of uncertainty quantification for graph neural operators, first focusing on a second-order elliptic PDE. Our primary aim is to demonstrate how Bayesian graph neural operators can identify regions of uncertainty in solution estimates and mitigate prediction errors in low-sampling regimes.

To recreate the results in Li et al. (2020b) we first use their original code for graph-based neural operators[1] using message-passing layers (Kipf & Welling, 2016; Gilmer et al., 2017) with 64 hidden dimensions and ReLU activations. Training is performed via the Adam optimizer. We then apply our last-layer Laplace approximation (as outlined in Section 3.2) post hoc, via the library introduced by Daxberger et al. (2021). This method constructs a full generalized Gauss–Newton approximation (Schraudolph, 2002) of the Hessian of the training loss at the final-layer weights. Two scalar hyperparameters—the prior precision and observation

---

[1]https://github.com/zongyi-li/graph-pde/graph-neural-operator

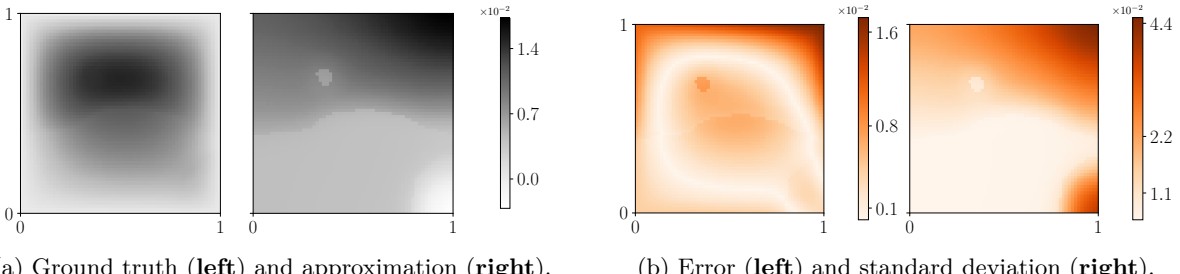

(a) Ground truth (**left**) and approximation (**right**).   (b) Error (**left**) and standard deviation (**right**).

Figure 4: The Bayesian neural operator applied to the 2D Darcy flow problem in a low-data regime. The approximation is poor, and the predictive standard deviation highlights the areas of high error.

noise—are tuned *post hoc* by optimizing the log marginal likelihood (Immer et al., 2021; Daxberger et al., 2021).

We consider the second-order elliptic PDE examined in Li et al. (2020b), given by

$$-\nabla \cdot (\lambda(x)\nabla u(x)) = f(x), \quad x \in D$$
$$u(x) = 0 \qquad x \in \partial D \tag{21}$$

where $D = [0,1]^2$ is the unit square and $f \equiv 1$. The PDE in Equation (21) represents the steady state of a two dimensional Darcy flow and arises in several physical applications. Note that even though the PDE is linear, the parameter-to-solution operator $\lambda \mapsto u$ is not. The neural operator architecture approximates this operator via a graph-based neural network (Kipf & Welling (2016)). In particular, for the computation of the integral in Equation (11), the domain $D$ is discretised into a graph-structured data on which the message passing algorithm of Gilmer et al. (2017) is applied. In Section 5.2 we examine the case where only few data are available, while Section 5.2 addresses a high data regime.

**Low-data Regime** We begin by examining the case of sparse observation points on the unit square $D = [0,1]^2$ , a common scenario in multi-scale dynamics described by PDEs, where data is often expensive to obtain. In such cases, the limited data can lead to inaccurate approximations, making it essential to quantify the uncertainty associated with predictions.

In particular, since the problem is relatively simple, we consider an extreme setting where we train on only two training functions and subsample only two points from a $16 \times 16$ grid for each. Figure 4 shows on a $61 \times 61$ grid that in this setting the NO fails to predict the solution well. As a consequence, our method exhibits low confidence (high predictive standard deviation) in the prediction, particularly in the areas of higher error. For readability, the plots use different color scales. This is due to the slight underconfidence of the Laplace approximation (in the scalar global parameter, not the local structure). Having measures such as the predictive standard deviation to determine whether the prediction should be trusted is of big practical benefit for many applications.

**High-data Regime** The previous section examined a heavily under-sampled scenario, characterized by a limited amount of training data. While this setup may appear simplified, under-sampling is a common challenge in practical applications involving high-dimensional problems, where it is often infeasible to densely sample the domain with pre-computed PDE solutions. In this section, for completeness, we explore the opposite end of the spectrum—a highly over-sampled regime—and find that good and structured uncertainty quantification is nevertheless useful here.

Figure 5 shows results on a dense $61 \times 61$ grid, analogous to the previous one, trained on 100 densely evaluated $16 \times 16$ grid solutions. Note, that the model generalizes well from the smaller $16 \times 16$ grid used during training to the larger $61 \times 61$ grid for testing, as previously shown by Li et al. (2020b). Although the prediction error is generally of good quality (i.e. relative prediction errors are mostly below 10%), the trained network exhibits an artifact in one, sharply delineated region of the training domain. This is a common problem with the ReLU features in this architecture, which create piecewise linear predictive regions (Hein

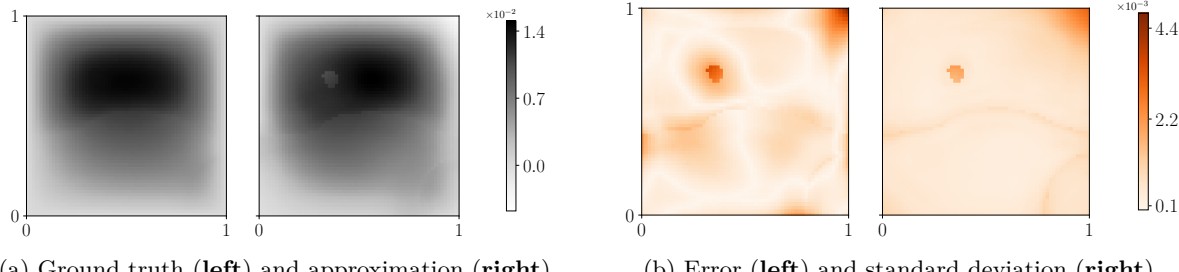

(a) Ground truth (**left**) and approximation (**right**).   (b) Error (**left**) and standard deviation (**right**).

Figure 5: The Bayesian neural operator on the 2d Darcy flow problem in the high-data regime. The approximation is close to the ground truth. The regions of relatively high error, as well as their magnitude, are captured by the predictive standard deviation.

et al., 2019). As the figure shows, the Laplace approximation is in fact able to identify and delineate this region well, and produce an effective, well-calibrated warning about its presence. It is important to note that this kind of functionality is only possible with the *structured* uncertainty produced by a Bayesian technique like the Laplace approximation – i.e. by an approximate posterior measure, rather than a global worst-case error bound.

## 5.3 Evaluation on APEbench

Having qualitatively illustrated the utility of our method, we now present a quantitative evaluation on a standardized benchmark of PDE problems. We assess uncertainty quantification performance on a diverse set of $1d$ equations from APEBench (Koehler et al., 2024), including Burgers', hyper-diffusion, and Kuramoto–Sivashinsky equations.

To benchmark our approach, we compare it against three widely used uncertainty estimation baselines in deep learning:

- *Deep ensembles* (Lakshminarayanan et al., 2017; Hansen & Salamon, 1990), which aggregate predictions from multiple independently trained models with different random initializations;
- *Input perturbations* (Pathak et al., 2022), which sample predictions by injecting noise into the input;
- *Weight perturbations*, which do so by perturbing the model weights.

The latter two methods approximate uncertainty by sampling multiple forward passes and then fitting a Gaussian distribution via empirical mean and covariance (moment matching). These approaches rely on the sensitivity of the model to input or parameter changes to reflect predictive uncertainty.

**Model and training.**   Our graph neural operator implementation has four layers with 18 hidden features each. The model takes the PDE's initial condition as input and lifts it to a hidden space via a linear projection. Each graph block: (i) identifies neighboring points within a fixed radius $r$; (ii) computes a learned kernel from the query–neighbor coordinates using a small MLP; (iii) aggregates neighbors via a sum to approximate the integral; and (iv) adds a linear term as (message or) a skip connection. All components are implemented in jax (Bradbury et al., 2018). We train for 100 epochs on 40 trajectories using mean squared error loss and Adam, with $r \approx 0.3$ on grids of 256 points.

**Evaluation.**   We evaluate models in an autoregressive setup: given the previous 10 time steps, the model predicts the next one. For the ensemble baseline, we train 10 independently initialized models. Evaluation is performed on 100 input–output pairs. To quantify uncertainty, we apply a last-layer Laplace approximation using the full generalized Gauss-Newton (GGN) approximation of the Hessian (Schraudolph, 2002). The prior precision is selected via grid search, optimizing for marginal negative log-likelihood (NLL). The Laplace approximation is applied to the weights of the final projection layer, which we linearize to produce the output

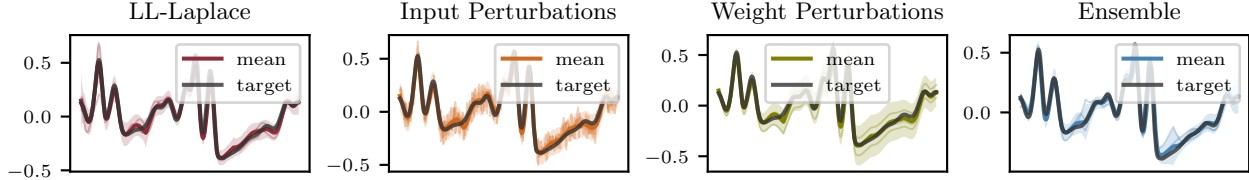

Figure 6: Predictive uncertainty of GNO for the $1d$ Korteweg–De Vries PDE under different UQ methods. We visualize the predictive mean, 1.96 standard deviation and samples.

predictive distribution. We report standard evaluation metrics: Root mean squared error (RMSE) of the predictive mean, expected marginal $\chi^2$-statistic (Q), and expected marginal negative log-likelihood (NLL):

$$\text{RMSE} = \sqrt{\frac{1}{n}\sum_{i=1}^{n}(u_i - \hat{u}_i)^2}, \quad \text{Q} = \frac{1}{n}\sum_{i=1}^{n}\frac{(u_i - \hat{u}_i)^2}{\sigma_i^2}, \quad \text{NLL} = -\sum_{i=1}^{n}\log\left(\frac{1}{\sqrt{2\pi\sigma_i^2}}\exp\left(-\frac{(u_i - \hat{u}_i)^2}{2\sigma_i^2}\right)\right).$$

Here, $u_i$ is the ground truth, $\hat{u}_i$ the predictive mean, and $\sigma_i$ the predicted standard deviation for the $i$-th test point. RMSE captures pointwise predictive accuracy; lower values indicate better performance. NLL quantifies how well the predicted distribution explains the data under the Gaussian assumption, with lower values indicating better calibration. The $\chi^2$-statistic assesses whether the predicted variances are appropriately scaled; values close to 1 suggest well-calibrated uncertainties.

The results show that our approach is generally competitive and often among the best in terms of uncertainty quantification. While we calibrated the prior precision via grid search, the observation noise (scaling the Gauss-Newton Hessian) was tuned manually. This value had a noticeable effect on Laplace performance, and ideally, both hyperparameters should be jointly optimized. Quantitative results for the Kuramoto–Sivashinsky and Korteweg–De Vries equations are shown in Table 1 and Table 2, with Figure 6 visualizing the predictive mean and uncertainty. Although deep ensembles sometimes match or exceed our method in predictive accuracy, they require training multiple independent models, which increases computational cost.

| Method | RMSE | Q | NLL |
|---|---|---|---|
| Input perturbations | 0.058 | 1.039 | $-1.421$ |
| Ensemble | 0.051 | 1.447 | $-1.521$ |
| Weight perturbations | 0.056 | 1.015 | $-1.376$ |
| LL-Laplace | 0.057 | 0.960 | $-1.349$ |

Table 1: Evaluation of UQ methods on the $1d$ Kuramoto-Sivashinsky equation.

| Method | RMSE | Q | NLL |
|---|---|---|---|
| Input perturbations | 0.050 | 0.762 | $-1.480$ |
| Ensemble | 0.066 | 1.556 | $-1.456$ |
| Weight perturbations | 0.052 | 0.796 | $-1.338$ |
| LL-Laplace | 0.049 | 0.667 | $-1.599$ |

Table 2: Evaluation of UQ methods on the $1d$ Korteweg–De Vries equation.

## 6 Conclusions

While neural operators have demonstrated competitive performance compared to other numerical methods and shown promise in outperforming neural network-based approaches on large grids for certain tasks, they do not come with explicit uncertainty quantification. We addressed this gap by developing an explicit analytic Bayesian treatment for the linear base-case, and illustrated how we can learn (the distribution over) solution operators through non-parametric GP regression. For the general deep setting, we focused on graph neural operators and proposed an efficient approximate Bayesian inference scheme based on Laplace approximations. Our experiments demonstrate that the proposed approach provides meaningful uncertainty estimates, both in sparse and dense data regimes.

If deep learning approaches to the simulation of dynamical systems are to fulfill their potential and be applied to serious, large-scale partial differential equations (including safety-critical and scientific applications), then uncertainty quantification as presented here has a crucial role to play in the prevention of accidental and potentially dangerous prediction errors.

## Acknowledgments

E.M. gratefully acknowledges financial support by the European Research Council through ERC CoG Action 101123955 ANUBIS ; the DFG Cluster of Excellence "Machine Learning - New Perspectives for Science", EXC 2064/1, project number 390727645; the German Federal Ministry of Education and Research (BMBF) through the Tübingen AI Center (FKZ: 01IS18039A); the DFG SPP 2298 (Project HE 7114/5-1), and the Carl Zeiss Foundation, (project "Certification and Foundations of Safe Machine Learning Systems in Healthcare"), as well as funds from the Ministry of Science, Research and Arts of the State of Baden-Württemberg. E.M. is grateful to the International Max Planck Research School for Intelligent Systems (IMPRS-IS) for support. N.K. was supported by a research grant (42062) from VILLUM FONDEN. N.K. acknowledges financial support by the Novo Nordisk Foundation through the Center for Basic Machine Learning Research in Life Science (NNF20OC0062606), and from the European Research Council (ERC) under the European Union's Horizon programme (grant agreement 101125993). L.R. acknowledges the financial support of: the European Commission (Horizon Europe grant ELIAS 101120237), the Ministry of Education, University and Research (FARE grant ML4IP R205T7J2KP) the European Research Council (grant SLING 819789), the US Air Force Office of Scientific Research (FA8655-22-1-7034), the Ministry of Education, the grant BAC FAIR PE00000013 funded by the EU - NGEU and the MIUR grant (PRIN 202244A7YL). This work represents only the view of the authors. The European Commission and the other organizations are not responsible for any use that may be made of the information it contains.

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

# A   Appendix

## A.1   Derivation of the Green's function for the one-dimensional Dirichlet problem

We consider the one-dimensional boundary value problem

$$\mathcal{L}_{\lambda_0} u\,(x) = \left(-\Delta - \lambda_0^2\,\mathrm{Id}\right)u(x) = \frac{d^2}{dx^2}u(x)\,-\,\lambda_0^2\,u(x) = f(x), \quad x \in [0,1],$$
$$u(0) = u(1) = 0, \tag{22}$$

The Green's function $G_{\lambda_0}(x,y)$ solves, for each $y \in [0,1]$,

$$\mathcal{L}_{\lambda_0}[\,G_{\lambda_0}(\,\cdot\,,y)\,](x) \;=\; \delta(x-y), \quad \text{with} \quad G_{\lambda_0}(0,y) \;=\; G_{\lambda_0}(1,y) \;=\; 0.$$

**Step 1: Solve the homogeneous equation away from $x = y$.**  For $x \neq y$, the Dirac delta is zero, so $G_{\lambda_0}$ satisfies the homogeneous problem

$$-\frac{d^2}{dx^2}G_{\lambda_0}(x,y) \;-\; \lambda_0^2\,G_{\lambda_0}(x,y) \;=\; 0.$$

Hence we can express the Green's function $G_{\lambda_0}$ in terms of the homogeneous equation solution. In the case of Equation (22) (considering also the boundary conditions) we have

$$G_{\lambda_0}(x,y) \;=\; \begin{cases} A(y)\,\sin\!\left(\lambda_0\,x\right), & 0 \leq x < y, \\ B(y)\,\sin\!\left(\lambda_0(1-x)\right), & y < x \leq 1. \end{cases} \tag{23}$$

where $A$ and $B$ might also depend on $\lambda_0$.

**Step 2: Enforce continuity at $x = y$.**  Since in this case $G_{\lambda_0}$ is continuous at $x = y$, we require

$$\lim_{x \to y^-} G_{\lambda_0}(x,y) \;=\; \lim_{x \to y^+} G_{\lambda_0}(x,y).$$

That is,

$$A\,\sin\!\left(\lambda_0\,y\right) \;=\; B\,\sin\!\left(\lambda_0(1-x)\right) \qquad \text{(Condition 1)}.$$

**Step 3: Impose the jump condition on the derivative.**  By integrating $\mathcal{L}_{\lambda_0}[\,G_{\lambda_0}(\,\cdot\,,y)\,](x) = \delta(x-y)$ across a small interval around $x = y$ we get

$$-\int_{y-\varepsilon}^{y+\varepsilon} G_{\lambda_0}''(x,y)\,dx \;=\; \int_{y-\varepsilon}^{y+\varepsilon} \delta(x-y)\,dx \;=\; 1.$$

Since

$$\int_{y-\varepsilon}^{y+\varepsilon} \frac{\partial}{\partial x^2}G_{\lambda_0}(x,y)\,dx \;=\; \frac{\partial}{\partial x}G_{\lambda_0}\big|_{x=y+\varepsilon} \;-\; \frac{\partial}{\partial x}G_{\lambda_0}\big|_{x=y-\varepsilon},$$

we get $-\left[\frac{\partial}{\partial x}G_{\lambda_0}\big|_{x=y+\varepsilon} - \frac{\partial}{\partial x}G_{\lambda_0}\big|_{x=y-\varepsilon}\right] = 1$, hence $\frac{\partial}{\partial x}G_{\lambda_0}\big|_{x=y+} - \frac{\partial}{\partial x}G_{\lambda_0}\big|_{x=y-} = -1$ For Equation (23) that is

$$-A\cos(\lambda_0 y) - B\cos\cos(\lambda_0(1-y)) = -\frac{1}{\lambda_0} \qquad \text{(Condition 2)}.$$

**Step 4: Solve for the coefficients.**  Solving for Conditions 1 and 2 leads to the known closed-form expression for $\lambda_0 \neq n\pi$:

$$G_{\lambda_0}(x,y) \;=\; \frac{1}{\lambda_0\,\sin(\lambda_0)} \begin{cases} \sin\!\left(\lambda_0\,x\right)\sin\!\left(\lambda_0\,(1-y)\right), & x \leq y, \\ \sin\!\left(\lambda_0\,y\right)\sin\!\left(\lambda_0\,(1-x)\right), & x \geq y. \end{cases}$$

One can equivalently write this piecewise definition in terms of the Heaviside step function $H$ (as in Equation (6)).

**Step 5: Verify the non-degeneracy condition.** If $\lambda_0 = n\pi$ for some $n \in \mathbb{N}$, then $\sin(\lambda_0) = 0$, and the above formula becomes singular. Indeed, in that case, the homogeneous problem with boundary conditions $u(0) = u(1) = 0$ has non-trivial solutions, which obstructs invertibility of $\mathcal{L}_{\lambda_0}$. Thus, the Green's function (and hence the unique solution) is well defined when $\lambda_0 \neq n\pi$.

This completes the derivation. For further details on Green's functions and partial differential equations, we refer e.g. to Stakgold & Holst (2011); Evans (2010); Olver et al. (2014). A derivation of this Green's function is also given in Skinner (2014).

## B   Additional results

Here we provide some additional experimental results.

| Method | RMSE | Q | NLL |
|---|---|---|---|
| Input perturbations | 0.047 | 0.802 | $-1.584$ |
| Ensemble | 0.045 | 0.899 | $-1.794$ |
| Weight perturbations | 0.047 | 0.860 | $-1.481$ |
| LL-Laplace | 0.047 | 0.861 | $-1.639$ |

Table 3: Evaluation of UQ methods on the $1d$ Burgers equation.

| Method | RMSE | Q | NLL |
|---|---|---|---|
| Input perturbations | 0.031 | 1.107 | $-2.022$ |
| Ensemble | 0.011 | 0.084 | $-2.364$ |
| Weight perturbations | 0.031 | 1.135 | $-1.972$ |
| LL-Laplace | 0.031 | 0.962 | $-2.021$ |

Table 4: Evaluation of UQ methods on the $1d$ hyperdiffusion equation.

| Method | RMSE | Q | NLL |
|---|---|---|---|
| Input perturbations | 0.049 | 0.992 | $-1.461$ |
| Ensemble | 0.059 | 0.679 | $-1.577$ |
| Weight perturbations | 0.047 | 0.796 | $-1.658$ |
| LL-Laplace | 0.049 | 0.826 | $-1.656$ |

Table 5: Evaluation of UQ methods on the $1d$ Fisher equation.

| Method | RMSE | Q | NLL |
|---|---|---|---|
| Input perturbations | 0.042 | 1.000 | $-1.656$ |
| Ensemble | 0.034 | 2.021 | $-2.356$ |
| Weight perturbations | 0.046 | 0.067 | $-0.431$ |
| LL-Laplace | 0.042 | 0.971 | $-1.830$ |

Table 6: Evaluation of UQ methods on the $1d$ nonlinear-diffusion equation.

