# OpenReview forum: "Approximate Bayesian Neural Operators: Uncertainty Quantification for Parametric PDEs"
_TMLR — Accepted by TMLR_

### Review · Reviewer_NH6G · 2024-12-18

**Summary Of Contributions:**

This paper considers uncertainty quantification of deep operator learning for PDE problems. The proposed method aims to learn the Green function of a linear PDE by a neural network, and provides posterior estimate of for the learned neural network with a Gaussian prior. The analytical results seem to be standard due the linearity of the problem and Gaussian assumption on the prior. The authors also extend the result to neural operators with multiple blocks (similar to FNO), by leveraging the fact the last layer is linear and by assuming the previous layer output is Gaussian.

**Audience:**

Yes

**Claims And Evidence:**

No

**Requested Changes:**

Summary:




Major issues:

The term “one-layer NN” throughout the whole paper is misleading. In the context of deep learning, one-layer NN usually refers to a NN with one hidden layer but the authors are referring to a sepcial neural operator (9) with one layer. Such neural network can be deep and contains many layers of activation and affine transforms.

Eqn (5), the author should provide more background (uniqueness and existence, and how the Green function is derived). A natural questions is if lambda_0 equals to square root of one of the eigenvalues of (-\Lambda) over the unit interval, this PDE should not admit a unique solution and the Green function representation fails.

The definition of NN in Section 2.2 is not clear and a lot of notations are used without defining it.
* The special form (11) requires that neural operators share the same kernel function in each layer but this is unnecessary nor practical. In particular, this is not consistent with the main reference Kovachki et al 2023. The author should justify such requirement and why it would be able to approximate a PDE operator.
* Eqn (11) suggests that the definition of g_\theta depends on a function \lambda. Is the \lambda here the same function in the PDE (1)? This is a bit confusing because a generic neural operator can approximate PDEs that are not in the form of (1). The author should justify why a special kernel in the form (11) is considered.
* W_l is not defined.

Section 2: “Loosely speaking, one can think of this construction as a deep neural network …” The discussion here is confusing. If author treats the output of each layer as a sequence that approximates the pde solution, similar to that of an iterative method, then why would using the exact same integral kernel g_\theta be helpful?

Discussion at the end of Section 2: how does UQ help predict PDE solutions when input functions are out of distribution cases?


The description of the proposed method needs a lot of work on clarification. The dependence of mean and covariance of (17) on the neural network is not clear. If no neural network is involved, this is simply a Bayesian posterior for a linear regression problem. Also (17) seems to be an important component of the proposed method, but the derivation and justification of (17) is missing.

End of section 3, extension from a single layer Neural Operator to multiple layer case is too brief and mathematical derivations are needed, especially when considering the numerical sections 5.2, 5.3 and 5.4 are all dedicated to this case. The author claims that the function ouput is Gaussian due to the linearity of the last layer, however this is only true when the previous layer output is Gaussian. Such assumption seems to be impractical due to the nonlinearity of neural networks.

Section 5, numerical setup for the neural operators are missing, e.g. the size and architecture of NO, as well as the training and tuning.

The authors only considers the uncertainty from prior distribution of the Green function. In my opinions, this is not the main challenges for uncertainty quantification for PDE operator learning, where the main uncertainty comes from the noises in training data, and stochastic optimization. Could you add a few comment on these problems?

Minor issues:

Figure 1 is mispositioned. It should be better placed in the same page as that of (6).

Discussions above (3), are the authors assuming (1) is a linear PDE, i.e. \mathcal{L}_\lambda is a linear differential operator? This seems to be an assumption through Section 2.

(4): the kernel should be G instead of G_\lambda.

(18): what does ``test functions” mean? u_\ast is not defined.



Typos:
Section 3, Page 5: Useable -> usable

(21): the covaraice has a subscript \Theta_{MAP}.

**Strengths And Weaknesses:**

Strength
* The problem of Uncertainty quantification for PDE operator learning is very important and this work helps provide insights toward this direction.
* The literatures on the neural network is clear and comprehensive.

Weakness
* The writing and description of the methods part are not clear and a lot of clarifications are needed, especially the extension to neural networks that are not "shallow". Please see more details in major and minor issues below.
* This paper seems to only consider a special neural network in the form (11), proposed by Li et al 2020. However, such architecture requires that every FNO block shares the same integration kernel function, which is not the case in generic FNO architecture, e.g. Kovachki 2023 and other references. Such assumption is too restrictive and the analysis conduct in this paper does not seem to rely on (11).
* The author seems to only consider the uncertainty from the Gaussian prior of the Green's function while in general, uncertainty of PDE operator learning usually comes from the random noises in training data and training process.

---

> ### Author Response · Authors · 2025-03-05
> **Response to comments and revisions**
>
> We thank the reviewer for the feedback and comments. Below we address the points raised:
>
> **Distinction between “Shallow” and “Deep” Case:**
>
> Thank you for raising this point. In our formulation, the shallow case is a special instance of the deep neural operator where $\lambda$ is fixed, $L=1$, $W_1=0$, and $\sigma$ set to the identity. When the differential operator is linear, this setting's integration kernel aligns with the Green's function, meaning the network effectively learns to invert the differential operator. For the deep case, each layer may be nonlinear, but our Bayesian treatment via Laplace approximation applies only to the last layer. The final layer's linearity ensures a Gaussian predictive distribution, assuming approximate Gaussianity in the preceding layer's output. To clarify this, we made the following changes:
>
> - We have now split the discussion into two distinct subsections: one for the shallow (linear) case (Equation (12)) and one for the deep (nonlinear) case, where multiple layers (each with its own integration kernel) are employed.
> - We added details to the extension to multiple layers. We explain how the Laplace approximation is applied to the final  layer while treating the preceding layers as a fixed feature extractor.
> - We also discuss how our method connects the Bayesian treatment of the shallow operator (via Gaussian processes) to that of deep neural operators.
>
> **Uncertainty from Data Noise:**
>
> The reviewer correctly notes that uncertainty in PDE operator learning also arises from data noise. In the revised manuscript, we have added a discussion that distinguishes between epistemic uncertainty in the model (captured by our Bayesian treatment over the network parameters) and data uncertainty (arising from noise in the training data). While our current focus is on model uncertainty via the Bayesian framework, we acknowledge that data noise is also an important component and outline potential extensions to incorporate input uncertainty through ensemble-based approaches in future work. Moreover, we address how this structured uncertainty can be leveraged to flag regions where the network is less confident—especially in the case of out-of-distribution inputs.
>
> Below we address point-by-point further revisions that we made:
>
> - “One-layer NN”: With on-layer NN we mean the shallow case described above ($\lambda$ fixed, $L=1$, $W_1=0$, and $\sigma=Id$ ). We have revised the text to avoid any confusion with conventional one-hidden-layer networks. The general deep case now allows different integration kernels for each layer.
> - Equation (5):  We have added the details for the conditions for uniqueness and existence of solutions.
> -  We defined $W_\ell$ as the local linear operators (matrices $\in \mathbb{R}^{d_{g_{l+1}}\times d_{g_{l}}}$) at each layer.
> -  In Equation (11), $\lambda$ is the parameter that defines the differential operator of the PDE (see Equation (1)). The general form of equation (1) allows $\mathcal{L}_\lambda$ to be any kind of differential operator –whether linear or nonlinear—making neural operators applicable to different PDEs. We have clarified this in the text.
> - We added clarification for equation (17).
> - We included more details on numerical setup, including size of neural operator architecture, tuning details etc.
> - We moved Figure 1 in the same page as that of (6).
> - In the revised version where we separate the shallow (linear) case and the deep (nonlinear approximated) case, it is made clear when the differential operator is assumed to be linear.
> - We corrected the typos, thank you for pointing them out.

---

### Review · Reviewer_Gc1F · 2024-12-27

**Summary Of Contributions:**

The paper develops an approximate Bayesian framework for neural operators to provide uncertainty estimates. First, an analytic Bayesian treatment through Gaussian processes is proposed for the linear, parametric PDEs. Then, an approximate Bayesian treatment through Laplace approximations is developed for the nonlinear PDEs.

**Audience:**

Yes

**Claims And Evidence:**

Yes

**Requested Changes:**

1.	More quantitative results are necessary to assess the quality of predictive uncertainty.

2.	It is suggested to compare this work with other related works.

3.	Existing experiments cannot fully support current expressions which gives an overstated application scope. More experiments are needed to support that the proposed approximate method is applicable to general deep neural operators. Otherwise, it is more suitable to say this work is targeted for a certain neural operator method.

**Strengths And Weaknesses:**

###  Strengths:

•  This paper provides an approximate Bayesian framework for neural operators for uncertainty quantification.

•  The paper is written clearly and easy to read.

•  The symbols and formulas in the paper can be clearly defined and expressed.

### Weaknesses:

•  This paper proposed two treatments for the linear PDEs (the “shallow” case) and nonlinear PDEs (the “deep” case) respectively. However, we do not know whether a new case is linear or nonlinear, so how can we determine which treatment to use?

•  In the experiments, this paper only gives the visualization results without any quantitative results.

•  As the authors summarized in related works, there are some works closely related to this work. It is highly recommended to compare this work with other related works and elaborate on the differences between this work and other related works.

•  The proposed approximate Bayesian framework is only validated on graph neural operators. Did the authors try the proposed Bayesian framework on other classical neural operator architectures (such as DeepONets and FNO) to prove the proposed method is basically universal?  If not, it seems overstated to say the proposed method applies to general neural operators. It is more suitable to say this work is targeted at a certain neural operator method.

---

> ### Author Response · Authors · 2025-03-05
> **Response to Reviewer Comments**
>
> We thank the reviewer for the feedback and comments. Below we address the points raised:
>
> **Shallow and deep case** :
> For linear PDEs with a closed-form Green’s function, our “shallow” treatment applies directly. For problems that do not have a closed-form solution or are nonlinear, we use the deep operator approach with a Laplace approximation on the last layer.
> The shallow case is primarily presented to build theoretical intuition, inspired by the Green’s solution formula for linear PDEs.
>
> **Graph Neural Operators**
> Our theoretical framework is formulated for general neural operators.
> In our experiments for the deep case, we use the graph neural operator architecture introduced in [1] because it is one of the earliest neural operator models. We have revised Sections 3.1 and 3.2 to clearly indicate that our experiments for the deep case are based on this specific architecture.
>
> **Experiments**:
> Our focus in this paper is on developing and illustrating the method—namely, using the Laplace approximation for uncertainty quantification in neural operators. The visualizations provided are meant to highlight the method’s potential. We agree that more quantitative results would further strengthen the work; however, extensive quantitative validation covering multiple different architectures and problems are planned for future work.
>
> **Related Work**: We have updated the section to clarify how our approach differs from the other related works.
>
> [1] Zongyi Li, Nikola Kovachki, Kamyar Azizzadenesheli, Burigede Liu, Kaushik Bhattacharya, Andrew Stuart, and Anima Anandkumar. Neural operator: Graph kernel network for partial differential equations. ICLR 2020 Workshop on Integration of Deep Neural Models and Differential Equations, 2020.

---

> > ### Comment · Reviewer_Gc1F · 2025-03-26
> > **Insufficient experiments.**
> >
> > In the revised version, the authors only distinguish their method from other related works literally but do not conduct the experiments to prove the advantages of their method over other methods, such as Magnani et al. (2024) which also employ Laplace approximations on FNO instead of GNO.
> >
> >
> > The authors only test a simple PDE, i.e., 2D Darcy flow. As other reviewers said, more numerical experiments applied to different PDEs are highly recommended to fully prove the advantages of the proposed method.

---

> > > ### Author Response · Authors · 2025-03-26
> > > **About the experiments**
> > >
> > > We agree that evaluating our approach on a broader range of PDEs is important, and we are actively working on extending the experimental evaluation in that direction. Unfortunately, due to unexpected external constraints unrelated to the manuscript, we were not able to include additional benchmarks beyond the 2D Darcy flow in time for the deadline on the current revision. However, as stated in the manuscript, this work specifically focuses on uncertainty quantification for graph neural operators, and we therefore limit our experiments to this class of architectures.

---

### Review · Reviewer_FdZ1 · 2025-02-21

**Summary Of Contributions:**

The authors present a framework for Bayesian uncertainty quantification for shallow (i.e., linear) neural operators. They consider this framework applied to deep neural networks via the Laplace approximation, i.e. applied to the final layer of the network. Finally, the authors include some numerical experiments to support their theory.

**Audience:**

No

**Broader Impact Concerns:**

No broader impact concerns.

**Claims And Evidence:**

Yes

**Requested Changes:**

Address in much greater the effect of using different meshes during training and evaluation. Describe how the GP framework for uncertainty estimates interacts (or doesn't) with mesh changes from a theoretical perspective in the shallow (linear) and deep (nonlinear) cases.

Add many numerical experiments applied to different PDE. This should include more experiments in the high-data and (especially) the low-data regime.

**Strengths And Weaknesses:**

Strengths:
  - The work has a nice, self-contained and accessible description of NOs, linear NOs, and their relationship to Greens functions of PDE.
  - The work convincingly makes the case for an important gap in the literature, namely UQ for NOs.

Weaknesses:
My criticisms have two main thrusts. The lack of treatment given to different meshes, and the sparseness of the numerical experiments.
  - The first is the lack of analysis/experiments on NOs with training data with _non-constant_ meshes. The main selling point of neural operators is their discretization independence. That is, that they allow for training/evaluation in different meshes. It isn't clear to me that the presented analysis generalizes to problems with multiple training/evaluation meshes, let alone if the analysis is discretization independent.
  - The authors restrict their analysis to NOs trained on a fixed mesh. I.e., they consider only one discretization. This, I believe, losses a lot of generality and means the main use case of NOs isn't covered.
  - What role does the grid play in the gaussian process being learned in the shallow (linear) case? What about the deep case with the Laplace approximation?
  - The numerical experiments are scant. Many more numerical experiments are needed to give confidence in the results. As it stands, there are two experiments that use the same PDE. This is simply not enough to give confidence that the results will generalize.
  - I also don't think the experimental results in figure 4 help make the case of the paper. The low-data case is important, but the approximate solution given is so low quality, and has such high error, it isn't clear that the reported uncertainty is meaningful.
  - Do the experimental results presented in figure 5 generalize when the data is not all given on the same mesh? I understand that the training and evaluation mesh are different, but what if the training mesh is not fixed?

---

> ### Author Response · Authors · 2025-03-05
> **Response to reviewer comments**
>
> We thank the reviewer for the feedback and comments. Below we address the points raised:
>
> **Discretization Independence**:
> Neural operators are inherently resolution invariant, enabling training and evaluation on different meshes. Our approach preserves this resolution-agnostic property by design, as the Laplace approximation models uncertainty over the network parameters rather than being tied to any specific discretization. Although this aspect was not the primary focus of our work, we appreciate the reviewer’s insight and will include a section in the revised manuscript to clarify how our framework maintains discretization independence.
>
> **Low-Data Regime and Quality of Uncertainty Estimates**:
> The low-data experiments were designed to illustrate that even when the approximate solution is poor, the uncertainty estimates accurately reflect the model’s lack of confidence. In such settings, we observe that the reported uncertainty is on the same scale as the actual error, thereby serving as a reliable indicator of when the neural operator is likely to be wrong. This is precisely where these estimates are most useful in practice.
>
> **Experimental Evaluation**:
> We agree that more quantitative results would strengthen the work. However, the primary goal of this paper is to introduce and illustrate the method—using the Laplace approximation for uncertainty quantification in neural operators—rather than to provide extensive quantitative benchmarks. The visualizations we provide are thought to show the method's potential. That said, we plan to conduct a broader quantitative evaluation, covering multiple architectures and problems.

---

> > ### Comment · Reviewer_FdZ1 · 2025-03-20
> > **Discretization Independence Section**
> >
> > You mentioned  that you would "include a section in the revised manuscript to clarify how our framework maintains discretization independence." Did this section make it into the newest revision?

---

> > > ### Author Response · Authors · 2025-03-20
> > > **Discretization Independence**
> > >
> > > We have now added a small section at the end of the methods section addressing how our approach inherits the mesh independence property. This aspect is also mentioned in the background section on neural operators.

---

### Author Response · Authors · 2025-02-27
**Acknowledgment of Reviews – Revisions in Progress**

Thank you very much for your detailed and constructive feedback. We acknowledge receipt of all reviews and will provide a detailed response soon to address the points raised.

---

### Comment · Action_Editor_fXpw · 2025-04-06
**not good enough**

This paper presents a Bayesian approach for uncertainty quantifization of neural operators. It starts with (shallow) linear case, which results in a GP, and then move to the nonlinear case via laplace approximation. Results were presented to demonstrate that the Bayesian neural operators (BNO) work as expected. Though uncertainty quantifizaiton is an important topic for neural operators, I agree with Reviewer FdZ1 and Reviewer NH6G that the current manuscript is not mature enough, espeically on the evaluation, which only demonstrates how BNO works, while not providing any evidence on the advantage of BNO over other methods. Thus, I recommend to reject the current manuscript and let the authors to substantially improve for a solid publication.

---

### Decision · Action_Editor_zukk · 2025-06-03

**Recommendation:** Accept with minor revision

**Additional Comments:**

Reviewers appreciated the direction of extending NOs to be uncertainty-aware, and I agree with them that this is an important and interesting direction. At the same time, in their reviews and during the rebuttal, some concerns about the depth of the experimental part were raised, e.g., by reviewers FdZ1 and NH6G. They main argument was that the manuscript demonstrated only how BNO works, while not providing any evidence on the advantage of BNO over other methods.

The authors uploaded a (late) revised version that includes more experiments and where the last layer Laplace approximation they propose for turning NOs into BNOs is compared against other baselines such as deep ensembles. After some careful consideration, I believe that this set of experiments is enough to address the reviewer's concerns and therefore I deem the paper ready to be accepted after a minor revision in which I ask the authors to briefly discuss other Bayesian approaches to PDE solution surrogates for UQ, such as applying the very same Laplace approximation to PINNs instead of NOs [a].

[a] - Beltran, Christian Jimenez, et al. "Galerkin meets Laplace: Fast uncertainty estimation in neural pdes." ICLR 2024 Workshop on AI4DifferentialEquations In Science. 2024.

**Audience:**

Yes

**Audience Explanation:**

UQ is definitely an interesting topic for the TMLR audience, as well as its application to neural operators and PDE surrogates in general.

**Claims And Evidence:**

Yes

**Claims Explanation:**

**disclaimer** I entered the review process in the last step as an emergency AE. I carefully read the paper, the reviews and their rebuttal as well as the new revision posted by the authors.

In this work, authors look at neural operators from a Bayesian lens, as they are interested in uncertainty quantification (UQ). They analyze Bayesian neural operators (BNO) by first  investigating the case of the "shallow" operators by bridging them to Gaussian processes and then in the regime of deep BNOs using classical methods from the approximate Bayesian inference toolkit such as the Laplace approximation. The resulting BNOs are tested on a number of (simple) PDEs and showed to yield meaningful uncertainty quantification.